# Trusted Aggregation (TAG): Model Filtering Backdoor Defense In Federated Learning

## Abstract

Federated Learning is a framework for training machine learning models from multiple local data sets without access to the data. A shared model is jointly learned through an interactive process between server and clients that combines locally learned model gradients or weights. However, the lack of data transparency naturally raises concerns about model security. Recently, several state-of-the-art backdoor attacks have been proposed, which achieve high attack success rates while simultaneously being difficult to detect, leading to compromised federated learning models. In this paper, motivated by differences in the output layer distribution between models trained with and without the presence of backdoor attacks, we propose a defense method that can prevent backdoor attacks from influencing the model while maintaining the accuracy of the original classification task.

## 1 Introduction

Federated learning (FL) is a potential solution to constructing a machine learning model from several local data sources that cannot be exchanged or aggregated. As mentioned in [8], these restrictions are essential in areas where data privacy or security is critical, including but not limited to healthcare. Also, FL is valuable for companies that shift computing workloads to local devices. Furthermore, these local data sets are not required to be independent and identically distributed. Hence, a shared robust global model is desirable and, in many cases, cannot be produced without some form of collaborative learning. Under the FL setting, local entities (clients) submit their locally learned model gradients and weights to be intelligently combined by some centralized entity (server) to create a shared and robust machine learning model.

Concerns have arisen that the lack of control or knowledge regarding the local training procedure could allow a user, with malicious intent, to create an update that compromises the global model for all participating clients. An example of such harm is a backdoor attack, where the malicious users try to get the global model to associate a given manipulation of the input data, known as a trigger, with a particular outcome. Some methods [6, 10, 7] have been proposed to detect the triggers in the training data to defend against backdoor attacks. However, in FL, as only the resulting model gradients or weights are communicated back, such methods cannot be applied to defend against backdoor attacks. Furthermore, since the model update in FL assumes no access to all clients' data, there is less information available to help detect and prevent such malicious intent. Thus backdoor attacks may be easier to perform and harder to detect in FL. Furthermore, current robust aggregation methods [15] fail to prevent even mild backdoor attacks.

In this paper, we first find that the output layer distributions of malicious users are very different from that of benign users. Specifically, there exists a discernible difference between malicious and benign user distributions for the target label class. Therefore, we can leverage this difference to detect backdoor attacks. Figure 1 shows a model with different estimated distributions for the target class depending on whether or not that model has been backdoor attacked.

Motivated by the finding that the output layer distributions of a model with and without a backdoor are different, we propose distributional differences between the output layers of returning user models and a known clean model to identify malicious updates. The proposed method is effective against multiple state-of-the-art backdoor attacks at different strength levels. Even in the unreasonable setting where 40% of the clients are malicious for each update, we greatly delay the success of the backdoor attack, outperforming current robust aggregation methods. In the experiment section, we demonstrate our method's ability on several data sets to prevent backdoor attacks. The method performs well even when the attack happens every round starting at the beginning of the process. Furthermore, our method does not affect the performance of the global model on clean data, resulting in no decrease and even increases in the accuracy of the original classification task.

## 2 Related Work

**Federated Learning.** Federated learning (FL) is an emerging machine learning paradigm that has seen great success in many fields [11, 3, 1]. At a high level, FL is an iterative procedure involving rounds of model improvement until it meets some criteria. These rounds send the global model to users and select a subset of users to update the global model. Then those chosen users train their local copy of the model, and their resulting models are communicated back and aggregated to create a new global model. Typically, the final local model's gradients or weights are transmitted back to ensure data privacy. Popular aggregation methods of FL include FedAvg [9], Median [16] and Trim-mean [16].

**Backdoor Attack.** Recently, several backdoor attacks have been proposed to take advantage of the FL setting. In [14], the authors show that the multiple-user nature of FL can be exploitable to make more potent and lasting backdoor attacks. By distributing the backdoor trigger across a few malicious users, they could make the global model exhibit the desired behavior at higher rates and for many iterations after the attack had concluded. We will show our threshold's effectiveness in even more potent attack settings than in their original paper.

A recent work [17] proposed a projection method, Neurotoxin, for any backdoor attack method to improve the longevity of the compromise to a model. The attacker's updates are projected onto dimensions with small absolute values of the weight vector. The authors claim such weights are updated less frequently by other benign users, resulting in greater longevity of successful attacks. We will demonstrate our method's effectiveness against both of the above attacks [14, 17].

**Defense.** On the other hand, few defense methods have been proposed to defend against backdoor attacks in FL. Prior work [12] claims that norm clipping [13] is effective against backdoor attacks in FL but has been broken by the Neurotoxin attack. Two other robust defense methods for FL were proposed in [15]. The paper theoretically explores two robust aggregation methods: Median and Trim-mean, which were shown effective in defending against poisoning attacks in FL. Median is a coordinate-wise aggregation rule in which the aggregated weight vector is generated by computing the coordinate-wise median among the weight vectors of selected users. Trim-mean aggregates the weight vectors by computing the coordinate-wise mean using trimmed values, meaning that each dimension's top and bottom $k$ elements will not be used. We propose a method that can be implemented in addition to other aggregation or model filtering methods. In the experiment, we focus on the original FedAvg [9] aggregation to show the effectiveness of our proposed method without assistance from additional defense techniques.

## 3 Method

This section describes the motivation and framework for our proposed method, Trusted Aggregation (TAG), which effectively defends against state-of-the-art backdoor attacks. The current defense aggregation methods [9, 15] are insufficient for preventing attacks of even mild strength. In addition to better model security, our method can improve accuracy for the original classification task compared to the current robust aggregation methods.

**Motivation.** We find that the output layer distributions of models returned by malicious users are very different from that of benign users. Figure 1 shows the output distributions of a backdoor model and a clean model on clean input data. Each neuron in the output layer corresponds to one class, and the backdoor model has a learned association between the backdoor trigger and the target class. We observe that the learned associated comes with a distributional change in the output distribution for the target class. Therefore it implies that with a guaranteed clean model, we should be able to identify whether another candidate model has a backdoor attack by comparing their output distributions on

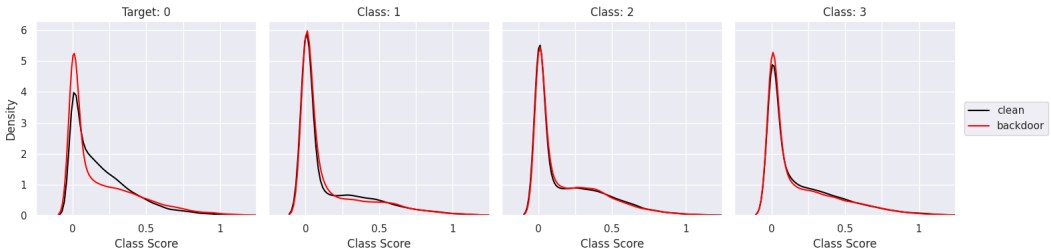

Figure 1: Final hidden layer output distributions (kernel density estimation based) for a **backdoor** model (red) and a **clean** model (black). There is an obvious difference between the distributions of the backdoor and clean models for the target label class.

some clean data. Note that we can observe a discernible difference between malicious and benign user distributions for this target label class. Therefore, we can leverage this difference to detect backdoor attacks.

**Detection Framework.**   We assume that there exists one user who we can be confident is trustworthy to place in charge of gate-keeping the global model for updates. The detection method leverages the trusted user to evaluate incoming model weights and determine whether each contribution is allowed to participate in the global model update procedure. The assumption is reasonable as, in reality, the center server will also collect some data to help with the training process, not just blindly relying on the local data from users.

The main idea is to detect user models with an unusually distributed output layer with information from a single trusted user. Moving forward, we will refer to this single trusted user as the validation user. In each communication round, this validation user completes the following steps to generate a threshold for malicious user detection, see Algorithm 1

---

**Algorithm 1** Trusted Aggregation

Notation: Let $\boldsymbol{S}$ represent the random subset of users that will submit locally trained models $U_j$ to update the global model $G$, $U_T$ to denote the model from the trusted user, $\boldsymbol{X}$ to denote the local data of the trusted user, and $\mathcal{D}$ to represent the distributional difference function.

---

1: **procedure** TRUSTED AGGREGATION($\boldsymbol{X}, G, U_T, \{U_j\}_{j \in \boldsymbol{S}}$)
2:     Generated outputs: $\boldsymbol{o}_G = G(\boldsymbol{X})$, $\boldsymbol{o}_T = U_T(\boldsymbol{X})$, and $\boldsymbol{o}_j = U_j(\boldsymbol{X}), \forall j \in \boldsymbol{S}$
3:     **for** each class $c \in [1, ..., m]$ **do**
4:         Compute the distributional distances between each user and the global model
5:             $v_T^{(c)} = \mathcal{D}(\boldsymbol{o}_G^{(c)}, \boldsymbol{o}_T^{(c)})$ and $v_j^{(c)} = \mathcal{D}(\boldsymbol{o}_G^{(c)}, \boldsymbol{o}_j^{(c)}), \forall j \in \boldsymbol{S}$         ▷ $\boldsymbol{o}^{(c)}$: output for class $c$
6:     **end for**
7:     The above procedure produces: $\boldsymbol{v}_T \in \mathbb{R}^m, \boldsymbol{v}_j \in \mathbb{R}^m$         ▷ $m$: total number of classes
8:     Compute threshold: $\tau = 2 \times \max(\boldsymbol{v}_T)$         ▷ max: maximum element of the vector
9:     $\tilde{\tau} \leftarrow$ GLOBAL-MIN MEAN SMOOTHING($\tau$)         ▷ Algorithm 2
10:     Select users: $\boldsymbol{S}_r = \{j \in \boldsymbol{S} \,|\, \max(\boldsymbol{v}_j) < \tilde{\tau}\}$         ▷ maximum element < threshold
11:     **return** FedAvg($\{U_j\}_{j \in \boldsymbol{S}_r}$)
12: **end procedure**

---

In general, Algorithm 1 determines which users will be used for the global updates based on a threshold. During each round of training, we compute and store a forward pass output ($\boldsymbol{o}_G$) of the global model on the validation user's local data. Then, local training is performed, and forward pass outputs ($\boldsymbol{o}_j, \boldsymbol{o}_T$) on the validation user's local data with the selected users' models and the model of the validation user are stored. For each class, we compute the class-conditional distributional distance ($v_j^{(c)}, v_T^{(c)}$) between the global model output ($\boldsymbol{o}_G^{(c)}$) and the user output ($\boldsymbol{o}_j^{(c)}$ or $\boldsymbol{o}_T^{(c)}$) by applying a distributional difference function on estimated CDFs based on $\boldsymbol{o}_G^{(c)}, \boldsymbol{o}_j^{(c)}$ and $\boldsymbol{o}_T^{(c)}$. Here, $\boldsymbol{o}^{(c)}$ represents the outputs based on the trusted user's local data with the label $c$. In our experiment, the Kolmogorov-Smirnov (KS) function is used to compute the distributional difference, but other distance functions can also be applied. Suppose there are $m$ classes in total; the process will result in a distance vector ($\boldsymbol{v}_j, \boldsymbol{v}_T \in \mathbb{R}^m$) for each user, including the validation user. The distance vectors will then determine which users can be selected for the update.

**Threshold Construction.** In this part, we discuss how to decide the threshold ($\tau$) and how to use it to select users. We quantify the threshold as the largest possible change a non-malicious user could contribute. Users with distance values exceeding the threshold will be excluded. Assume that the class-conditional distances ($v^{(c)}$) are Uniform on $[0, b_c]$ for each class $c$, where $b_c$ is the maximum possible change to the output layer of class $c$ through local training by a non-malicious user. Therefore, the threshold can be generated by estimating the maximum of $b_c$ for any class. Let $m$ represent the total number of classes, equation 1 shows that under the assumption, twice the maximum of the class-conditional distance ($2\max(v^{(c)})$) is a practical estimation of the upper bound of $b_c, \forall c \in [1, ..., m]$.

$$\forall c \in [1, ..., m], v^{(c)} \sim \text{Uniform}(0, b_c), \text{let } j = \arg\max_c (b_c) \text{ such that } b_j = \max_c (b_c).$$

$$\max_c \left( v^{(c)} \right) \geq v^{(j)} \implies E\left[\max_c \left(v^{(c)}\right)\right] \geq E\left[v^{(j)}\right] = \frac{b_j}{2} \implies E\left[2 \times \max_c \left(v^{(c)}\right)\right] \geq b_j \quad (1)$$

Since the validation user is non-malicious, their distance vector serves as a good representation for other non-malicious users. Therefore, we estimate the threshold $\tau$ by setting $\tau = 2 \times \max(\boldsymbol{v}_T)$, where $\boldsymbol{v}_T \in \mathbb{R}^m$ is the distance vector of the validation user and $\max(\cdot)$ means getting the maximum value of the vector $\boldsymbol{v}_T$. Then, the maximum distance value ($\max(\boldsymbol{v}_j)$) of each selected user will be compared with the threshold ($\tau$) to determine the final list of users who can participate in the update. A user with a maximum distance smaller than the threshold is considered a benign user, while a user with a maximum distance larger than or equal to the threshold will be removed. However, this naive threshold is very unstable, and a lucky malicious user can get past it in some rounds due to the instability. Therefore, we make an additional modification, **global-min mean smoothing**, to this basic threshold to address the concern.

**Global-Min Mean Smoothing.** A straightforward way to stabilize the threshold value is smoothing methods. However, in the early communication rounds, the naive threshold value rapidly decreases as the model starts making connections between inputs and output classes. Therefore, applying a smoothing method early will result in a relatively high threshold, which may let attackers bypass it. When the naive threshold ($\tau$) decreases rapidly, we do not wish to use any previous communication rounds for the smoothing.

Therefore, we propose to use the lowest observed value (Global Min) of $\tau$ as the starting point of smoothing. Let $\tau_t$ represent the naive estimation of the threshold in round $t$, the smoothed threshold $\tilde{\tau}$ at round $n$ is given by

$$\tilde{\tau} = \frac{1}{n - t_s + 1} \sum_{t=t_s}^{n} \tau_t,$$

where $t_s$ is the round that when the global min is observed. Details of the global-min mean smoothing is described in Algorithm 2. As $\tau_t$ shrinks, we observe new global minimums, and the start of the threshold smoothing is reset. In addition, when our estimate stabilizes, previous values are leveraged to smooth the threshold, which keeps lucky malicious users from getting past a volatile threshold. Figure 2 compares our global min-mean smoothing with the naive threshold and various smoothing techniques. The

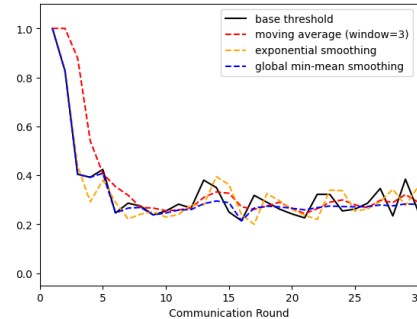

Figure 2: Comparison of the global min-mean smoothing with the base (naive) threshold and various smoothing methods.

global min-mean smoothing best captures the naive threshold's early behavior while providing remarkable stability improvements. Additionally, when our threshold encounters a new global minimum, it provides a conservative estimate to prevent malicious users while re-learning cutoff behavior over the next few rounds.

---

**Algorithm 2** Global-Min Mean Smoothing

Notation: Let $(\tau_1, \cdots, \tau_{n-1}, \tau_n)$ denote the sequence of values that we wish to smooth.

1: **procedure** GLOBAL-MIN MEAN SMOOTHING($\tau_1, \cdots, \tau_{n-1}, \tau_n$)
2:      Record the location of global minimum: $i = \arg\min_{t \in [1,...,n]} \tau_t$
3:      Subset to a sequence starting with the global min: $\{\tau_t\}_{t=i}^n = \{\tau_i, \cdots, \tau_n\}$
4:      **return** average of sequence subset, $\overline{\{\tau_t\}_{t=i}^n}$
5: **end procedure**

---

## 4  EXPERIMENTS

### 4.1  Setting

**Federated Learning.**  We start by giving further specifications regarding the federated learning environment. Our interest is training a global model over $M$ communication rounds with $N$ users. Each iteration randomly selects $K$ users, using a specified proportion of the total users, to participate in the model update. After local training, the next global model is the average returned model weights by the FedAvg procedure. We focus our experiments on the ResNet18 model architecture; a standard object recognition classifier initially proposed in [4]. We assume that all users, including malicious, have complete control over all aspects of local training, such as learning rate, the number of epochs, and the model weights they return. For simplicity, we select two main sets of training hyper-parameters for benign and malicious users. The malicious users will poison a given proportion of their local data by adding their backdoor trigger to the input and changing the training label to the target class. They intend for the model to associate the trigger with the target class and hence have the future global model identify any input with the trigger as belonging to the target class.

**Attack and Baseline.**  To show the effectiveness of our method, we choose a setting in which the backdoor attack is strong. We force all malicious users to be included in the subset of selected users to update the global model each round after the start of the backdoor attack. Note that the selection of random users is a defense against malicious users by making it difficult for them to update the global model repeatedly. Additionally, we do not allow the validation user, a guaranteed benign user, to participate in any global model updates. We make these decisions to show the ability of our threshold to prevent even strong backdoor attacks against the global model. For our experiment, we test the proposed method and two other robust aggregation methods, Median and Trim-mean [15], against two state-of-the-art backdoor attacks in FL: Neurotoxin [17] and Distributed Backdoor Attacks (DBA) [14]. To further evaluate the effectiveness of the aggregation methods, we also vary the proportion of malicious attackers ($10\%, 40\%$) in selected users to test the defense methods under different attack strength levels.

**Data.**  The experiments are done on three different data sets: CIFAR10 [5], STL10 [2] and CIFAR100 [5]. In each experiment, we randomly split the data between the users. For global model evaluation, we split the test set into two parts. We add the backdoor trigger to images in the second half and remove any target class observations. We measure model performance with classification accuracy using the first half as classification accuracy, and the proportion of the poisoned half predicted as the target class, known as attack success rate, to measure the extent that the backdoor attack has compromised the model. For a defense method, a good performance consists of a low attack success rate and high classification accuracy. In other words, both attacks are unsuccessful when the defense method is used, and the defense does not negatively influence the classification performance.

### 4.2  Results

We begin by considering a setting where 10% of the selected users is malicious each communication round. Figure 3 shows the performance of the three robust aggregation methods against DBA and Neurotoxin attacks on three data sets regarding classification accuracy and attack success rate. Our proposed method (TAG) nullifies the backdoor attack in each case without decreasing the classification accuracy of the original task. Furthermore, the model reaches a clear improvement in the model's classification accuracy on the CIFAR-10 data set compared to the other two aggregation methods. The other two robust aggregation methods, coordinate-wise Median and Trim-mean, only prevent the backdoor attack on STL10 with Neurotoxin. We conclude that our method is a clear improvement to the existing robust aggregation methods for federated learning.

We show that TAG can handle even stronger attack settings against state-of-the-art attacks in the following part. We consider testing the robust aggregation methods against DBA and Neurotoxin attacks with 40% malicious users in the selected set. These attacks are catastrophically successful against the current robust aggregation methods, see Figure 4, having a nearly perfect attack success rate after round 50 on all our data sets. However, our method, TAG, overcomes the backdoor extent of the initial rounds to prevent the attack against both CIFAR data sets. Although our defense method eventually could not withstand the Neurotoxin attack on STL10, we note that incredibly TAG delayed the attack's success for nearly 90 communication rounds when nearly half of the users were malicious. TAG's performance against DBA on STL10 is also unsatisfactory, but it still delays the attack's success.

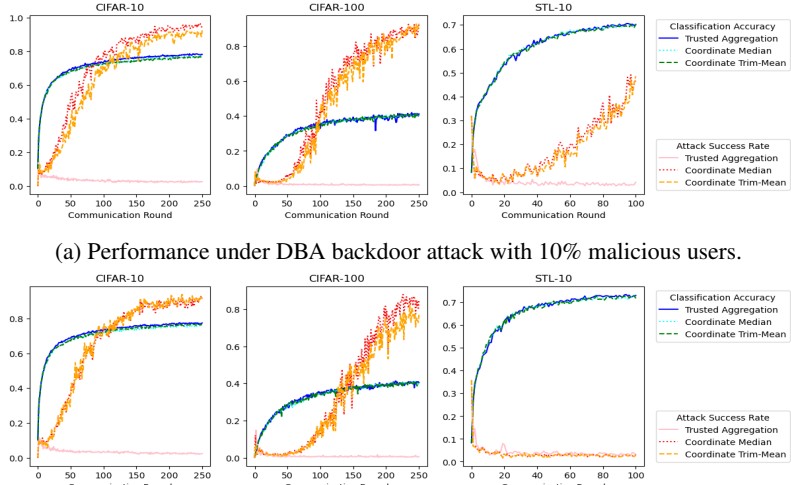

(a) Performance under DBA backdoor attack with 10% malicious users.

(b) Performance under DBA and Neurotoxin backdoor attacks with 10% malicious users.

Figure 3: Model performance under DBA and Neurotoxin backdoor attacks with 10% malicious users. The proposed method TAG performs well in defending against backdoor attacks as the attack success rates are low. Meanwhile, it does not affect the model's classification performance on clean data. However, the other two aggregations methods do not work well against backdoor attacks except on STL10 against Neurotoxin.

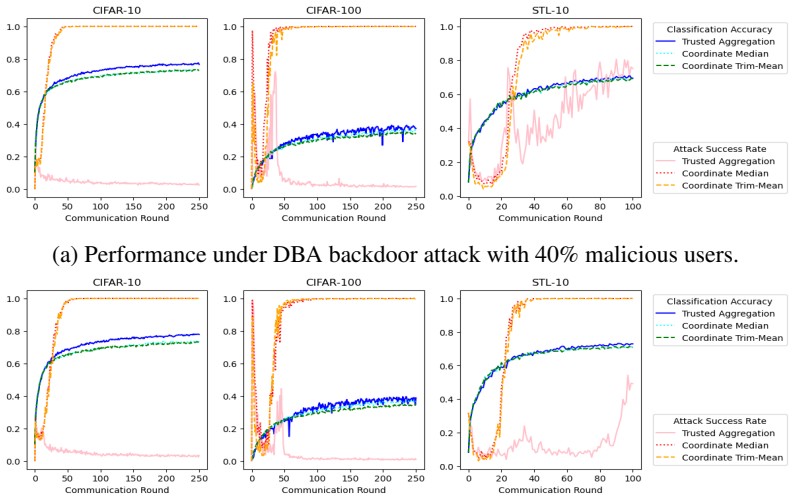

(a) Performance under DBA backdoor attack with 40% malicious users.

(b) Performance under DBA and Neurotoxin backdoor attacks with 40% malicious users.

Figure 4: Model performance under DBA and Neurotoxin backdoor attacks with 40% malicious users. The proposed method TAG performs well in defending against the backdoor attacks on CIFAR10 and CIFAR100. However, the other two aggregation methods do not work well on the three data sets.

## 5 Conclusion

We believe our proposed method, Trusted Aggregation (TAG), is an essential advancement toward model security for the federated learning framework. While current robust aggregation methods fail to prevent mild backdoor attacks, TAG holds up against state-of-the-art attacks in unreasonably strong settings. Furthermore, TAG can act as a layer of model filtering in addition to current and future modifications to the choice of aggregation step.

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
