# OpenReview forum: "Trusted Aggregation (TAG): Model Filtering Backdoor Defense In Federated Learning"
_NeurIPS.cc/2022/Workshop/Federated_Learning — FL-NeurIPS 2022 Poster_

### Official Review · Reviewer_bLkT · 2022-10-05
**Collaborative Defense to Backdoors**

The proposed algorithm is based on an empirical observation that the output logits of a compromised model have a different form than the benign ones. Although, this observation can be potentially true, it doesn't really cover cases when non-iid client distributions (quite frequent in real-life FL scenarios). My understanding might be wrong however there isn't any theoretical proof for this basic assumption. I need something more substantial than Fig. 1 to convince me that this core assumption holds.

Besides this basic assumption, the algorithm has a few more drawbacks:
- It removes some clients from the aggregation: As such, their contribution to the global model is eliminated and this can lead to under-representation, especially in case of non-iid distributions.
- Besides the unfairness mentioned above, the process looks problematic for the non-iid case where the models can produce very different responses (based on their local data).
- There is a weak link is the trusted client that might introduce artifacts in the entire process.
- Finally, the notion of privacy is severely breached since all models need to be evaluated on a single client's data. As such, either the logistics or the fact that all models are now used on a single client are incomprehensible.

A few more comments about the experiments are:
- an ablation study is necessary to understand how many clients are included per iteration. Similarly, it would be nice to know how the ratio of malicious users affects the overall performance (although it seems that it doesn't affect much according to Figures 3 and 4).
- what is the contribution of the two steps in the overall performance? I would like to understand what exactly is the contribution of the two components of the proposed algorithm
- what would happen in the non-iid case? I would like to see similar experimental results for the case of label mismatch across the clients

Based on all these comments, my recommendation is to marginally reject the paper. However, the idea is interesting and some additional analysis and elaboration can go a long way with this paper

---

### Official Review · Reviewer_nMgR · 2022-10-14
**Proposed an interesting robust aggregation algorithm but more thorough analysis is needed**

In this paper, the authors proposed a filter-type robust aggregation algorithm for defense against backdoor attacks in federated learning. In particular, they assume that there is a trusted party that has access to clean data, which will be used to compute a threshold to filter out possible malicious parties’ replies. An additional module named Global-Min Mean Smoothing is also needed to strengthen the computation of the threshold.

* Main issues
1. he authors does not seem to consider or mention the non-iid case where the parties may have non-iid local data distribution. How does this scenario affect the hidden layer output distribution?
2. In the numerical experiments, the authors did not mention how many parties are there in the considered FL system, and how is the data distributed?
3. Again in the numerical experiment results, the authors does not provide ablation results on how important is the Global-Min Mean Smoothing module, and how much clean data a trusted party needs to have. Do those factors affect the performance of the proposed algorithm?

* Minor Typos

Fig 3b) and 4b) have wrong captions?

---

### Official Review · Reviewer_ggjb · 2022-10-18
**an interesting defense against backdoor attack**

In this paper, the authors introduce a model filtering backdoor defense for federated learning. The intuition comes from the observation that the resulting output layer distribution of a model from the backdoor attack is quite different from the honest model ones.
The defense is conducted by a trusted user who uses its own local data to filter out the potential other users’ models. This user could be considered as the central server with some public data. At every iteration, it calculates the distributional distances between the local model and the global model on the output layer. The trusted user’s distance is used to set-up the threshold and a smoothing strategy is proposed to stabilize the threshold.

The paper is well organized and easy to follow. The experimental results show the interest of such defense against the backdoor attack.

Few comments below:
•	In the experiments, it would be interested to see the ratio such that the defense has been successfully identified the malicious users, or the ratio such that it fails. We can then have a measure on the gap to pursue in the future.

•	In the setting of the experiments, the authors mentioned that the trusted user is not participated to every iteration. Is it true that more frequently it participates, better the defense will be? For the iterations that the trusted user does not participate, the models to filtered out are from the same users identified in the previous iterations?

•	For the threshold setup, it might be possible from the statistical point, to get an unbiased estimator of the $b_j$, e.g., a minimum-variance unbiased estimator. However, the trusted user needs to waits for a sufficient number of rounds to setup the threshold.  But this threshold might be more stable. I think there might be more justified ways to setup the threshold.

•	I know it might be hard but it would be interesting to show the statistical error rate of the defense. The strong point of median and trimmed mean method is that they have theoretical guarantee although they appear to be less efficient in the experiments. Besides, some literatures are missing which could be considered as benchmarks as well:
        Robust Aggregation for Federated Learning
        Byzantine Fault-Tolerant Distributed Machine Learning using D-SGD and Norm-Based Comparative Gradient Elimination (CGE)

---

### Decision · Program_Chairs · 2022-10-20

Accept (Poster)